# Depression and suicidal ideation among medical students in a private medical college of Bangladesh. A cross sectional web based survey

**Rifat Jahan Chomon**[ID]*

Department of Community Medicine, Enam Medical College and Hospital, Savar, Dhaka, Bangladesh

* chomonpath62@gmail.com

**Data Availability Statement:** All relevant data are within the paper and its Supporting information files.

## Abstract

This study was done to investigate the prevalence of depression and suicidal ideation among private medical students in Bangladesh. A total of 237 medical students participated in this cross-sectional web-based survey by e-questionnaire using the Google Form. The study was conducted from November 2020 to December 2020 at Enam Medical College and Hospital which is situated in Savar, Dhaka, Bangladesh. Out of 237 medical students, prevalence of depression was found 58.6%, and prevalence of suicidal ideation was found 27.4% which is higher than the global prevalence. Bivariate and multivariate analysis and logistic regression-based odds ratios (ORs) was done to see the association between grade of depression and suicidal thoughts with different variables. Association between grade of depression with family problems ($x^2$ = 16.716, P = 0.001), drug addiction ($x^2$ = 16.601, P = 0.001), committed relationship status ($x^2$ = 40.292, P = <0.001) were statistically significant. Whereas, the association between suicidal thoughts with family problems ($x^2$ = 29.881, P = <0.001), failed any subject during study ($x^2$ = 12.024, P = 0.007), alcohol uses ($x^2$ = 15.977, P = 0.001), drug addiction ($x^2$ = 22.633, P = <0.001), committed relationship status ($x^2$ = 35.219, P = <0.001) were statistically significant. However, medical students whom had to earn other than their family income were 2.3 times (OR: 2.285, 95% CL: 0.897, 5.820) greater prone to be depressed than those who do not had to earn by themselves. On the contrary, medical students who are single were 2.35 times (OR: 2.352, 95% CL: 0.926, 5.973) greater prone to have suicidal thoughts than the married students. This study showed that a large percentage of Bangladeshi medical students have been suffering from depression and suicidal ideation. Our recommendation for the authority of the medical colleges are, to build a system with counselling facilities inside every medical colleges in Bangladesh.

## Introduction

Depression is at every corner of the earth. Doctors, workers, students are all susceptible to this intense feeling of dread. Now depression isn't a linear feeling. Depression is complex, there's

**Funding:** The author received no specific funding for this work.

**Competing interests:** The authors have declared that no competing interests exist.

multiple emotions to it, multiple reasons and quite a lot of different things can cause a person to be depressed. Studying medicine to be a doctor is a very noble deed but with it comes a hefty amount of stress. Medical students have quite a lot of responsibility and on top of that there are multiple courses that really strain the mind. Each course requires a lot of attention and hard work. On top of these courses, the grades are based on quite a lot of tests and oral exams. All this stress makes a lot of medical students quit. Seeing dropouts is a common thing nowadays.

The Covid-19 pandemic has been a real pain for medical students as well. Having classes canceled, delaying professional exams have added to the extreme onset of stress that they are already facing. This delay in their studies and not having a slight clue when their studies will continue properly is a huge factor when it comes to stress [1].

As per WHO, depression is the second-greatest predominant mental situation in the world. There is a plethora of proof that show us how academic stress, anxiety, and depression are higher in medical students, and consequently, these proportions keep on getting higher when these students become a doctor and start taking responsibility for other people's lives [2].

With the increase of depression among students, suicide is starting to become a real problem. It has been recognized as the fifteenth biggest reason for death around the world, that too among 15 to 29 year olds. Suicide takes almost 10,000 lives a year and also affects a lot of the youth of Bangladesh [3]. To add on to the issue, studies have found that most of these young people are underdiagnosed and don't get the proper treatment either, and the number of these suicide, depression, and mood disorders are increasing day by day [4].

Prior studies haven't really shown the huge problem of depression in medical students. Late meta-analysis and recent systematic reviews shows prevalence of depression amongst medical students was about 27.2% around the world, and for suicidal ideation prevalence was 11.1% [5].

While medical colleges in other developed nations are utilizing a total confirmation measure that takes persuasive expositions, test scores, and organized meetings into account, Bangladesh on the other hand, enlists students according to what they score in the college placement tests.

In China, a total of 39 studies was done between 1997 and 2015 including 32,694 university students. The overall prevalence of depression was found 23.8% [6]. In Italian, a cross-sectional study performed in 12 medical schools, and the prevalence of depression was found 29.5% [7]. In Korea, a study shows medical students had a greater depression percentage (52.3%) than students of engineering (34.0%) [8]. Another examination in Indian amongst 1st year medical students shows that the anxiety rate was generally much more predominant [9].

A study done in the United Kingdom indicated that more than 33% of first-year students had poor psychological wellness, as per the General Health Questionnaire 12, that evaluates depression and anxiety [10]. In Saudi Arabia, another examination shows that the prevalence of depression and anxiety of all types amongst medical students of a medical college was around 57% [11].

Medical students are really susceptible to suicide and the reasons vary a lot. Individual and professional distress, no relaxation time, money related obligation, homesickness, academic burden, and work pressure [12]. A study takes a gander at the degree of suicidal ideation in Bangladesh, uncovers the most noteworthy pace of suicide occurred amongst the age of 16 and 20 years, and family strife is well thought-out as the main explanation behind suicide in Bangladesh [13].

A medical student means one more set of hands that will treat a sick person and will most probably directly mean another life saved, losing them to suicide is a sad misfortune. But sad to say, the suicide rate among is on the rise [14]. As Bangladesh is an emerging nation, there is

a possibility that the outdated education system might cause a lot of stress to the students [15]. And with that, there is a much larger issue that comes on about, a larger public health issue that plagues the world [16].

Assessing the symptoms of depression, and suicidal tendencies is the first step to actually fixing this problem in. Hence, it is important for us to re-investigate the factors that are prevalent among these medical students which are inadvertently affecting their mental state.

## Methods and materials

### Study period, design and participants

The study was carried out from November 2020 to December 2020 in Enam Medical College & Hospital, Savar, Dhaka, Bangladesh. A web-based cross-sectional study design was used by e-questionnaire using Google form. Stratified Random Sampling technique was applied in this study. 1st year to 5th year students were the total population. Each year was consider as a strata (i.e., 1st year is a strata). From each of the strata 45 samples was collected randomly (Fig 1).

### Sample size and sampling procedure

Single population proportion formula was used to calculate the sample size.

$$n = Z^2 pq/d^2$$

Where, n = Sample size, z = level of confidence according to the standard normal distribution (for a level of confidence of 95%, z = 1.96), p = estimated proportion of the population that presents the characteristic (Here we use p = 0.82), d = tolerated margin of error (for example we want to know the real proportion within 5%), q = (1 − p). Thus, placing all values, **n = $(1.96)^2$ (0.82) (0.18)/ $(0.05)^2$ = 226**. So total 226 respondents was participating in this study.

### Data collection tools and technique

A web-based, self-administered and structured questionnaire was conducted to collect the data. The scale Beck Depression Inventory II (BDI-II) was used to measure depression and suicidal ideation among private medical students. This scale is chosen because it has questions about suicidal thoughts. Beck Depression Inventory II scale was aimed to document a variety of depressive symptoms the individual experienced over the previous week. It is a 21-item measure which are made on a 4-point scale. It is ranging from 0 to 3. The total scores can range from 0 to 63. The questionnaire was developed in English.

### Data management and analysis plan

Data was collected by e-questionnaire and managed via Microsoft excel. The statistical analysis of the data was carried out by using the software program SPSS version 20. Data was coded, checked, cleaned, and edited properly before analysis. Bi-variate, multivariate, and logistic regression based Odds Ratio analyses were done to find out the association between status of depression and suicidal ideation with various factors. Odds Ratio at 95% Confidence Interval (CI) was calculated, and p-value of different variables were measured.

### Data quality control

A pre-test was done on 5% of the total sample in Enam Medical College & Hospital and necessary adjustments were made. During data collection process, the questionnaire was checked for its completeness on daily basis by the supervisors and the investigators.

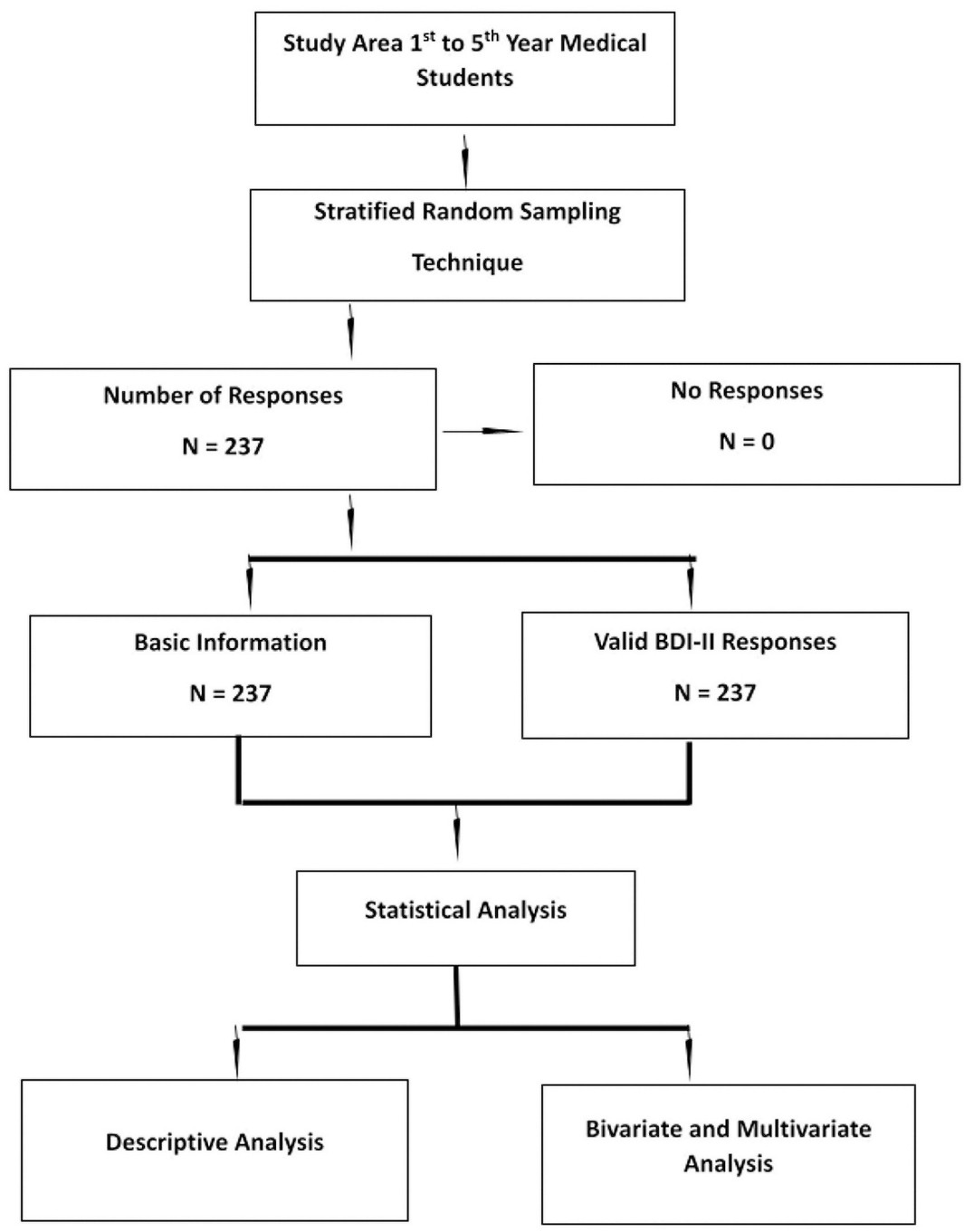

**Fig 1. Flow chart.**

## Ethical consideration

Formal requests was made to the appropriate authority for getting permission to collect data from a joint ethical review committee of the Enam Medical College & Hospital. Written informed consent was taken from the respondents prior to the data collection. Those who did

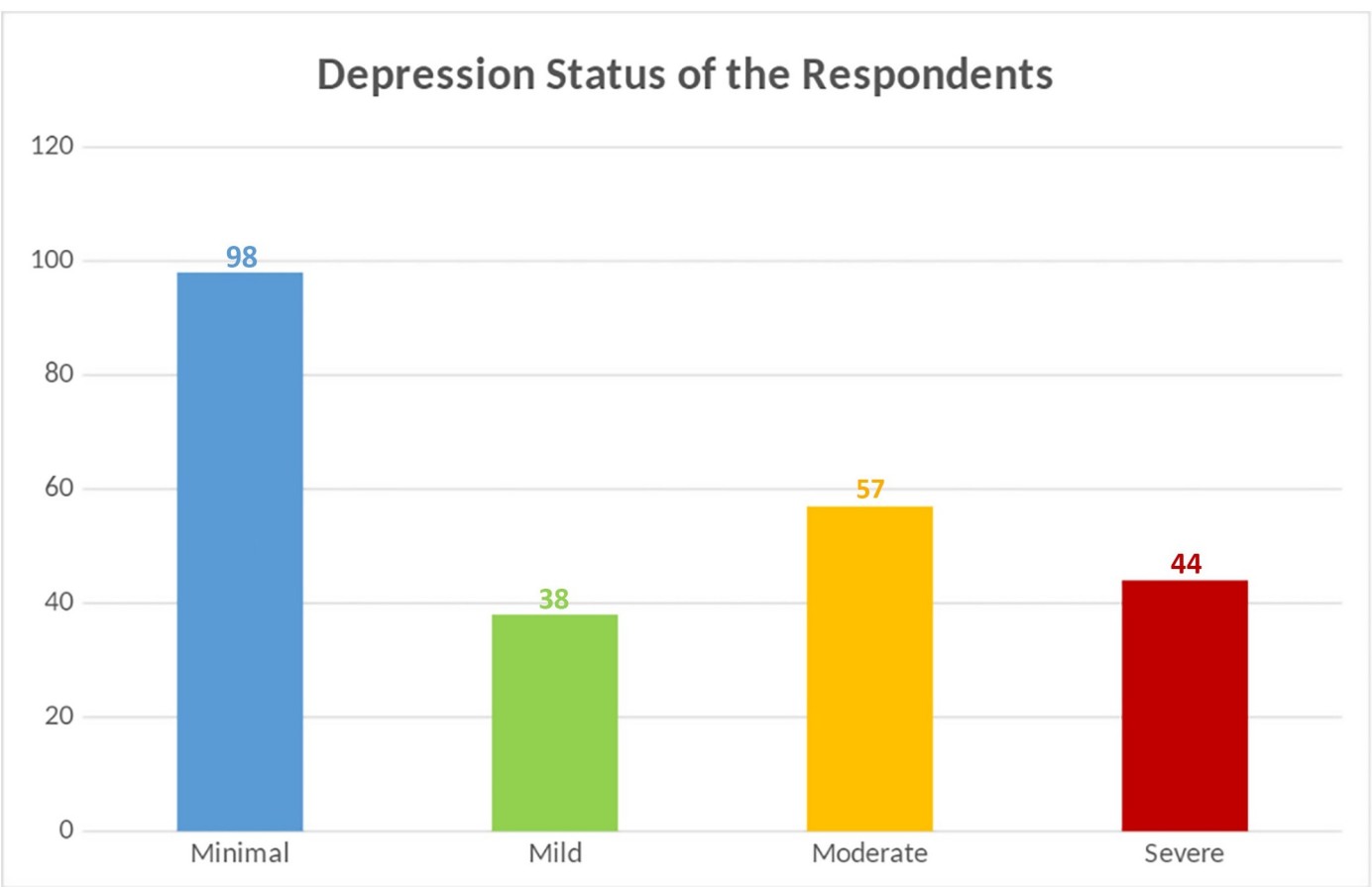

**Fig 2. Depression status of the respondents.**

not want to take part in the study were allowed not to participate or to withdraw from the study at any time they want.

## Results

### Depression status of the respondents

Out of 237 medical students participated in the study, the overall prevalence was found 58.6%. Among those with depression a majority 139 participants had mild to severe depressive signs. Rest 98 (41%) participants were found no depression or had minimal depressive symptoms (Fig 2).

### Descriptive data of various designated variables of the medical students

In this study, 118 (49.8%) students were male and rest of the students were female. 35% students have family problems and more the 21.1% students have family history of depression. Approximately half of the students (42.2%) were found that they have failed in any subject during their study period. 5.9% students had history of taking drugs. Less than a quarter percent of students (22.4%) had history of committed relationship broken up in their lifetime (Table 1).

**Table 1. Frequency and percentage of different variables among medical students of Enam Medical College & Hospital, Savar, Dhaka, Bangladesh, 2020 (n = 237).**

| Variables | Frequency | Percent |
|---|---|---|
| Gender | | |
| Male | 118 | 49.8 |
| Female | 119 | 50.2 |
| Age | | |
| 16–19 years | 10 | 4.2 |
| 20–23 years | 168 | 70.9 |
| 24 years and above | 59 | 24.9 |
| Marital Status | | |
| Single | 217 | 91.6 |
| Married | 19 | 8.0 |
| Divorce | 1 | 0.4 |
| Family Type | | |
| Adjunct Family | 214 | 90.3 |
| Broken Family | 23 | 9.7 |
| Family Problems | | |
| Present | 83 | 35.0 |
| Absent | 154 | 65.0 |
| Family History of Depression | | |
| Present | 50 | 21.1 |
| Absent | 187 | 78.9 |
| Earing other than guardian | | |
| Yes | 20 | 8.4 |
| No | 217 | 91.6 |
| Year of Studying | | |
| 1st | 46 | 19.4 |
| 2nd | 47 | 19.8 |
| 3rd | 46 | 19.4 |
| 4th | 48 | 20.3 |
| 5th | 50 | 21.1 |
| Monthly Family Income | | |
| 10000–40000 Taka | 62 | 26.2 |
| 41000–80000 Taka | 81 | 34.2 |
| 81000–120000 Taka | 58 | 24.5 |
| Above 120000 Taka | 36 | 15.2 |
| Failed any subject during study | | |
| Yes | 100 | 42.2 |
| No | 137 | 57.8 |
| Alcohol Uses | | |
| Everyday | 1 | 0.4 |
| Twice a week | 1 | 0.4 |
| Once a week | 4 | 1.7 |
| None | 231 | 97.5 |
| Drug Addiction | | |
| Oral | 6 | 2.5 |
| Inhale | 7 | 3.0 |
| Inject | 1 | 0.4 |

(*Continued*)

**Table 1.** (Continued)

| Variables | Frequency | Percent |
|---|---|---|
| None | 223 | 94.1 |
| Duration of committed relation broke up | | |
| Within a month | 9 | 3.8 |
| Three month or earlier | 44 | 18.6 |
| I am in a good relation | 69 | 29.1 |
| I never had any relation | 115 | 48.5 |

## Suicidal thoughts level of all the participants

It appears that 172 (72.57%) students from medical college do not have any suicidal thoughts at all and they are passing their normal life. 51 (21.5%) students had thoughts of killing themselves, but do not carry them out and 5 (2.1%) students would like of kill themselves in their lifetime. About 9 (3.8%) students in this study were in the group of they would kill themselves if they had the chance (Fig 3).

## P-value of the different selected variables

The association between grade of depression with age ($x^2$ = 11.392, P = 0.010), family problems ($x^2$ = 16.716, P = 0.001), family history of depression ($x^2$ = 15.836, P = 0.001), year of studying

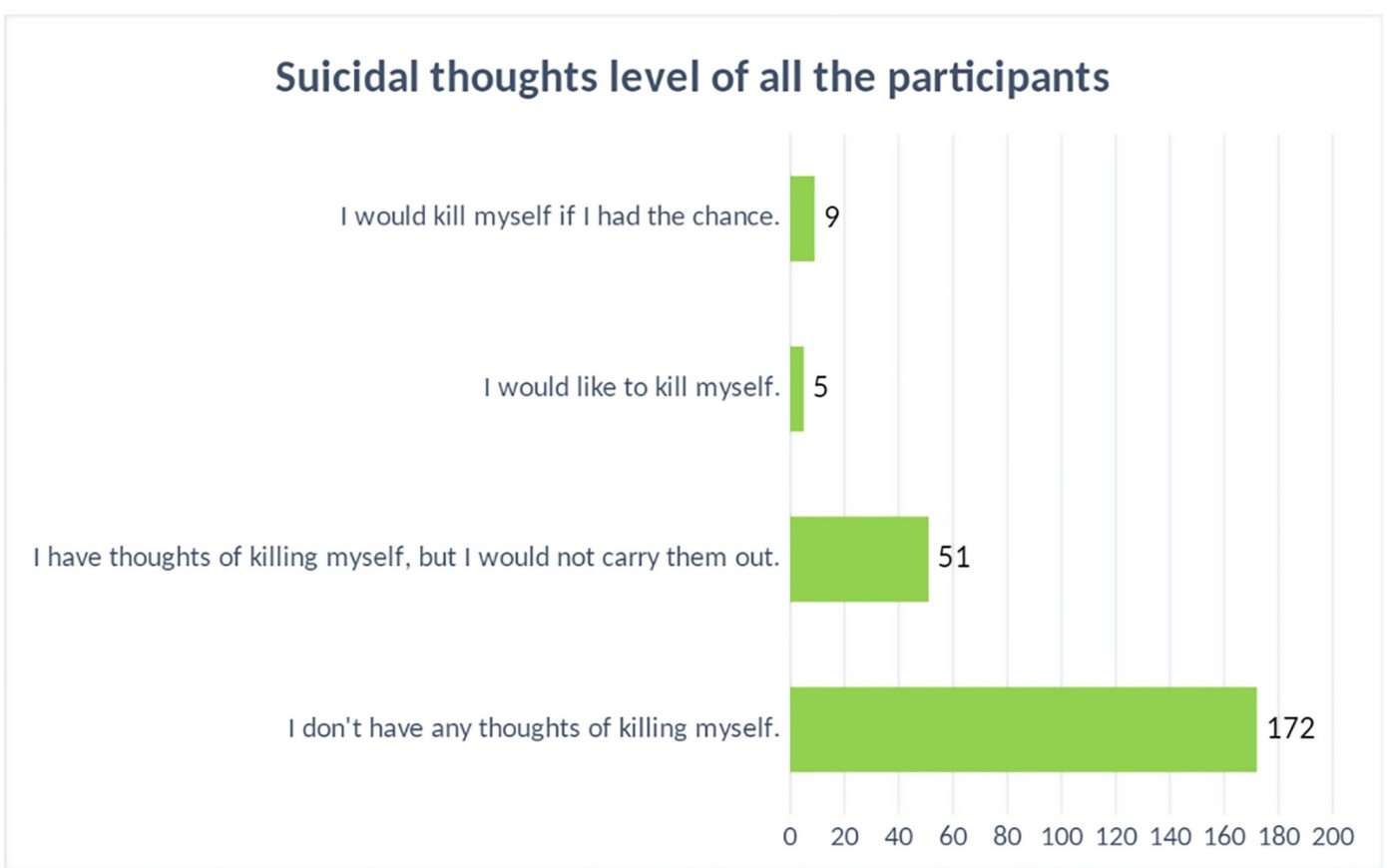

**Fig 3. Suicidal thoughts levels of all the participants.**

($x^2$ = 10.159, p = 0.017), drug addiction ($x^2$ = 16.601, P = 0.001), committed relationship status ($x^2$ = 40.292, P = <0.001) were statistically significant. On the other hand, the association between suicidal thoughts with marital status ($x^2$ = 9.711, P = 0.021), family problems ($x^2$ = 29.881, P = <0.001), family history of depression ($x^2$ = 12.010, P = 0.007), failed any subject during study ($x^2$ = 12.024, P = 0.007), alcohol uses ($x^2$ = 15.977, P = 0.001), drug addiction ($x^2$ = 22.633, P = <0.001), committed relationship status ($x^2$ = 35.219, P = <0.001) were statistically significant (Table 2).

## Odds ratio and 95% Confidence Interval of different variables

In this study, males were 1.08 times (OR: 1.088, 95% CL: 0.648, 1.824) greater prone to be depressed than the female. Students whose age between 16–23 years were 1.8 times (OR: 1.859, 95% CL: 0.992, 3.484) greater prone to be depressed than the students whose age between 24 years and above. Students who lives with the adjunct family were 1.7 times (OR: 1.691, 95% CL: 0.668, 4.280) greater prone to be depressed than the students who lives with broken family. Medical students whom had earning other than their family income were 2.3 times (OR: 2.285, 95% CL: 0.897, 5.820) greater prone to be depressed than those who do not had any earning. 1st, 2nd, and 3rd year medical students were 1.6 times (OR: 1.606, 95% CL: 0.942, 2.738) greater prone to be depressed than the 4th and 5th year students.

On the other hand, males were 1.45 times (OR: 1.450, 95% CL: 0.816, 2.577) greater prone to have suicidal thoughts than the female. Undergraduates whose age between 16–23 years were 1.68 times (OR: 1.683, 95% CL: 0.894, 3.168) greater prone to have suicidal thoughts than the undergraduates whose age between 24 years and above. Medical students who are single were 2.35 times (OR: 2.352, 95% CL: 0.926, 5.973) greater prone to have suicidal thoughts than the married students. Students who lives with the adjunct family were 2.2 times (OR: 2.224, 95% CL: 0.923, 5.359) greater prone to have suicidal thoughts than the students who lives with broken family. 1st, 2nd, and 3rd year medical students were 1.4 times (OR: 1.429, 95% CL: 0.804, 2.540) more likely to have suicidal thoughts than the 4th and 5th year students. Students whose family monthly income is 10000–80000 Taka were 1.1 times (OR: 1.114, 95% CL: 0.623, 1.990) greater prone to have suicidal thoughts than the students whose family income is 81000 and above (Table 3).

## Discussion

The COVID-19 pandemic has become an educational crisis for students worldwide [17]. In this devastating and challenging period, because of the continued lockdown students have developed different psychological problems like depression and suicidal thoughts [18]. This research is done to evaluate the prevalence and determinants of depressive symptoms, and suicidal tendencies in students of private medical colleges in Bangladesh during this pandemic. We also evaluated the socio-demographic, lifestyle related and health status related factors to find the association with depression and suicidal thoughts among the students.

In this research, the overall prevalence of depression was found to be 58.6%, and of suicidal ideation was found 27.4%, which is higher than the global prevalence of depression 27.2% and the prevalence of suicidal ideation was 11.1% among the medical students estimated in a systematic review and meta-analysis study [5]. In this study, findings of the prevalence were higher compared to the prevalence of 29.5% which was conducted in Italian medical students [7]. Vietnam reported that prevalence of depression was 15.2% and prevalence of suicidal thoughts were 7.7% which was conducted in a single medical school [19]. However, the prevalence of depression and suicidal tendencies differ all over the globe, researches have estimated higher prevalence of depression and suicidal ideation in a consistent manner. Several risk

**Table 2. Prevalence of depression and suicidal ideation according to associated factors among medical students of Enam Medical College & Hospital, Savar, Dhaka, Bangladesh, 2020 (n = 237).**

| Variables | Depression | | Suicidal thoughts | |
|---|---|---|---|---|
| | $X^2$ | P-value | $X^2$ | P-value |
| Gender | | | | |
| Male | 3.169 | 0.366 | 2.058 | 0.560 |
| Female | | | | |
| Age | | | | |
| 16–23 years | 11.392 | 0.010 | 6.036 | 0.110 |
| 24 years and above | | | | |
| Marital Status | | | | |
| Single | 3.874 | 0.275 | 9.711 | 0.021 |
| Married | | | | |
| Family Type | | | | |
| Adjunct Family | 4.841 | 0.184 | 5.133 | 0.162 |
| Broken Family | | | | |
| Family Problems | | | | |
| Present | 16.716 | 0.001 | 29.881 | <0.001 |
| Absent | | | | |
| Family History of Depression | | | | |
| Present | 15.836 | 0.001 | 12.010 | 0.007 |
| Absent | | | | |
| Earing other than guardian | | | | |
| Yes | 3.227 | 0.358 | 1.447 | 0.695 |
| No | | | | |
| Year of Studying | | | | |
| 1st, 2nd and 3rd | 10.159 | 0.017 | 6.082 | 0.108 |
| 4th and 5th | | | | |
| Monthly Family Income | | | | |
| 10000–80000 Taka | 2.367 | 0.500 | 0.991 | 0.804 |
| 81000 and above | | | | |
| Failed any subject during study | | | | |
| Yes | 7.591 | 0.055 | 12.024 | 0.007 |
| No | | | | |
| Alcohol Uses | | | | |
| Every day, twice a week or once a week | 4.499 | 0.212 | 15.977 | 0.001 |
| None | | | | |
| Drug Addiction | | | | |
| Oral, Inhale or Inject | 16.601 | 0.001 | 22.633 | <0.001 |
| None | | | | |
| Committed relationship status | | | | |
| Within a month or Three month or earlier | 40.292 | <0.001 | 35.219 | <0.001 |
| I am in a good relation or I never had any relation | | | | |

$X^2$ = Chi-square,

P-value less than 0.05 considered as significant

**Table 3. Odd Ratio and 95% Confidence Interval of different variables among medical students of Enam Medical College & Hospital, Savar, Dhaka, Bangladesh, 2020 (n = 237).**

| Variables | Depression | | Suicidal Thoughts | |
|---|---|---|---|---|
| | OR | 95% CL Lower-Upper | OR | 95% CL Lower-Upper |
| Gender | | | | |
| Male | 1.088 | 0.648–1.824 | 1.450 | 0.816–2.577 |
| Female | | | | |
| Age | | | | |
| 16–23 years | 1.859 | 0.992–3.484 | 1.683 | 0.894–3.168 |
| 24 years and above | | | | |
| Marital Status | | | | |
| Single | 0.548 | 0.218–1.376 | 2.352 | 0.926–5.973 |
| Married | | | | |
| Family Type | | | | |
| Adjunct Family | 1.691 | 0.668–4.280 | 2.224 | 0.923–5.359 |
| Broken Family | | | | |
| Family Problems | | | | |
| Present | 0.370 | 0.207–0.662 | 0.252 | 0.138–0.458 |
| Absent | | | | |
| Family History of Depression | | | | |
| Present | 0.422 | 0.211–0.844 | 0.380 | 0.198–0.731 |
| Absent | | | | |
| Earing other than guardian | | | | |
| Yes | 2.285 | 0.897–5.820 | 0.677 | 0.258–1.781 |
| No | | | | |
| Year of Studying | | | | |
| 1st, 2nd and 3rd | 1.606 | 0.942–2.738 | 1.429 | 0.804–2.540 |
| 4th and 5th | | | | |
| Monthly Family Income | | | | |
| 10000–80000 Taka | 0.857 | 0.506–1.452 | 1.114 | 0.623–1.990 |
| 81000 and above | | | | |
| Failed any subject during study | | | | |
| Yes | 0.588 | 0.345–1.001 | 0.568 | 0.319–1.010 |
| No | | | | |
| Alcohol Uses | | | | |
| Every day, twice a week or once a week | 0.703 | 0.126–3.917 | 0.179 | 0.032–1.004 |
| None | | | | |
| Drug Addiction | | | | |
| Oral, Inhale or Inject | 0.220 | 0.048–1.008 | 0.087 | 0.023–0.324 |
| None | | | | |
| Committed relationship status | | | | |
| Within a month and Three month or earlier | 0.254 | 0.120–0.535 | 0.180 | 0.093–0.347 |
| I am in a good relation and I never had any relation | | | | |

OR: Odd Ratio

CL: Confidence Interval

factors are responsible for this difference and these factors vary from country to country. In this research, it was found that a majority of the students have mild to severe depressive symptoms.

Prevalence rates of depression are estimated ranging from 15% to 82.4% and suicidal thoughts ranging from 1.8% to 53.6% in various studies. Among Chinese students, the suicidal tendencies were found to be around 15.24% [20]. Among Indian medical students it was found that more than half of the medical undergraduates are depressed [16]. In comparison, an investigation done in Saudi Arabia found that, the prevalence rate was 57% amongst medical undergraduates [11]. A different investigation in Karachi, Pakistan has shown that 35.6% of the medical undergraduates had suicidal ideation [14]. A study among 1st year medical students in United Kingdom showed the prevalence among students was 36% and almost 50% of the students depicted an unpleasant occurrence, most of which were related to medical training instead of individual issues [10]. Another study done in India among MBBS students showed us that 21.5% were have major depressive disorder which was measured by utilizing the Patient Health Questionnaire (PHQ-9) [9].

The current study has indicated the prevalence of depression is higher among medical students compared with the above studies. More than 3/4 of the depressed medical undergraduates had mild to severe level of depression in our study. Private medical college students in Bangladesh are experiencing a beyond compare growth of depression and suicidal ideation under the current global pandemic situation.

In this study, it was found that, age of the students, family problems, family history of depression, and year of studying, drug addiction, and committed relationship status had a significant association with the depression of medical students. On the other hand, the association between suicidal thoughts with marital status, family problems, family history of depression, failed any subject during study, alcohol uses, drug addiction, committed relationship status were statistically significant.

## Strengths and limitations

The strengths and limitations of the current investigation are dictated by a few issues. The questionnaire allows us to evaluate the prevalence of depression and suicidal thoughts among medical students while keeping up the WHO suggested 'social distance' during the COVID-19 pandemic, which in any case would be unimaginable. Additionally, the information for the study was gathered by worldwide approved normalized devices for quantitative investigation. In this cross-sectional study the different variables are viewed as associated factors, which could either be the causes or the consequences of depression or suicidal thoughts. Moreover, because of moral prerequisites on secrecy and privacy, the contact information of the respondents were not gathered. Since the exploration technique couldn't contact the individuals with medically analyzed depression side effects, the arrangement of the outcomes may not completely mirror the seriousness of depression and suicidal thoughts manifesting amongst medical students.

## Conclusion

Despite some limitations, this study gives empirical evidence that a large percentage of Bangladeshi medical students have been suffering from depression and suicidal thoughts. The prevalence of depression, and suicidal ideation among the medical students of Bangladesh is greater compared to other countries. Less than half of the students reported to have minimal depression. Majority of the students reported to suffer from mild to severe depression whereas one third of the students are suffering from suicidal tendencies.

## Recommendation

Our recommendation for the authority of the medical college are, to build a system with counselling facilities inside every medical college in Bangladesh. A Psychiatrist should be recruited for this counselling facilities with such demographics and therefore at a higher chance of suffering from depression to provide an in depth interview with these students and undertake a complete management according to the level of conditions of the students. However, routine examination of the students to detect any depressive symptoms among the current medical students inside the college should be included within the counselling facilities. Depression in medical students is also associated with some academic program, and therefore our recommendation is to detect the irregularities of those programs or departments and provide a solution for those irregularities as soon as possible and provide emergency counselling services for the students of those departments.

## Supporting information

**S1 Dataset. Data set used for analysis.**
(XLSX)

## Acknowledgments

We are grateful to the participants, as well as thankful to the editors and anonymous reviewers.

## Author Contributions

**Conceptualization:** Rifat Jahan Chomon.

**Formal analysis:** Rifat Jahan Chomon.

**Investigation:** Rifat Jahan Chomon.

**Methodology:** Rifat Jahan Chomon.

**Project administration:** Rifat Jahan Chomon.

**Supervision:** Rifat Jahan Chomon.

**Writing – original draft:** Rifat Jahan Chomon.

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
