## [Decision Letter · Decision Letter 0]

6 Jan 2022

PONE-D-21-39583Depression and suicidal ideation among medical students in a private medical college of Bangladesh. A cross sectional web based survey.PLOS ONE

Dear Dr. Chomon,

Thank you for submitting your manuscript to PLOS ONE. After careful consideration, we feel that it has merit but does not fully meet PLOS ONE’s publication criteria as it currently stands. Therefore, we invite you to submit a revised version of the manuscript that addresses the points raised during the review process.

We look forward to receiving your revised manuscript.

Kind regards,

Sanjay Kumar Singh Patel, Ph.D.

Academic Editor

PLOS ONE

Journal Requirements:

1 Please ensure that your manuscript meets PLOS ONE's style requirements, including those for file naming. The PLOS ONE style templates can be found at 

Reviewers' comments:

Reviewer's Responses to Questions

**Comments to the Author**

1. Is the manuscript technically sound, and do the data support the conclusions?

Reviewer #1: Yes

Reviewer #2: Yes

2. Has the statistical analysis been performed appropriately and rigorously? 

Reviewer #1: Yes

Reviewer #2: Yes

3. Have the authors made all data underlying the findings in their manuscript fully available?

Reviewer #1: Yes

Reviewer #2: Yes

4. Is the manuscript presented in an intelligible fashion and written in standard English?

Reviewer #1: Yes

Reviewer #2: No

5. Review Comments to the Author

Reviewer #1: Research article entitled "Depression and suicidal ideation among medical students in a private medical college of Bangladesh. A cross sectional web based survey. " focuses to investigate the prevalence of depression and suicidal ideation among private medical students in Bangladesh with 237 medical students participated in this cross-sectional web-based survey. Authors found prevalence of depression was found 58.6%, and prevalence of suicidal ideation was found 27.4% which is higher than the global prevalence. This study showed that a large percentage of Bangladeshi medical students have been suffering from depression and suicidal ideation. While the topic is of increasing relevance, still, this reviewer has certain suggestions that would help produce a more comprehensive study of the topic:

Specific comments that the author should consider

1, Figure 1 and 2 quality may be improved (high resolution).

2, It would be required to provide one illustrative Figure as to highlight the summary or future prospect of this study.

4, The English of manuscript can be polished (minor).

5, The authors should cross-check all abbreviations in the manuscript. Initially, define in full name followed by abbreviation.

6, Author should also consider COVID-19 pandemic in their study as this may be the added factor in depression.

7, Due to the COVID-19 pandemic work pressure (on medical fraternity) was another factor to be considered in this study.

Reviewer #2: In this paper entitled "Depression and suicidal ideation among medical students in a private medical college of Bangladesh. A cross-sectional web-based survey.," the author investigates the prevalence of depression and suicidal ideation among private students in Bangladesh. An online google form is used to perform the crosssectional survey in the manuscript. The study is well carried out and easy to understand. However, This manuscript requires a revision before its publication in PLOS One as follows:

Comments:

1) The English of the manuscript may be improved.

2) The current data only provide a glimpse of the private college in Bangladesh. The author has not considered including other medical colleges in the region for data collection. State reason?.

4) Limited statistical measure was used in the manuscript. Why has the author not used other statistical measures for data analysis, including logistic regression and others?.

5) The author has cited low-impact journals in the manuscript. However, many studies regarding depression and suicidal ideation are available on PubMed in prestigious journals like PLoS one and others.

6) Introduction: The importance of this study may be more highlighted explicitly at the start.

7) Discussion: The author may add a general discussion section to interpret the results with the literature or theories. The 

8) Improve the quality of the figures. The resolution and quality of the paper are low for publication.

---

## [Author Response · Author response to Decision Letter 0]

18 Feb 2022

I have been asked to explain why my data set only contains Enam Medical College students; it is because in our country, due to the pandemic, most colleges have been closed. As I am a faculty member of Enam Medical College, I was able to collect information by directly contacting the students through email and other online platforms. 

Minimal data set has been added as supporting information. One illustrative figure is added and the quality of the figures have been improved. Data analysis by logistic regression based Odds Ratio have been calculated. High-impact factor journals from PubMed and PLOS One were cited. The introduction and discussion parts have been improved to explain the importance of this study.

---

## [Decision Letter · Decision Letter 1]

1 Mar 2022

Depression and suicidal ideation among medical students in a private medical college of Bangladesh. A cross sectional web based survey.

PONE-D-21-39583R1

Dear Dr. Chomon,

We’re pleased to inform you that your manuscript has been judged scientifically suitable for publication and will be formally accepted for publication once it meets all outstanding technical requirements.

Kind regards,

Sanjay Kumar Singh Patel, Ph.D.

Academic Editor

PLOS ONE

Reviewers' comments:

Reviewer's Responses to Que

Reviewer #1: The manuscript entitled "Depression and suicidal ideation among medical students in a private medical college of Bangladesh. A cross sectional web based survey." has been improved form its first draft.

---

## [Editor Report · Acceptance letter]

3 Mar 2022

PONE-D-21-39583R1 

Depression and suicidal ideation among medical students in a private medical college of Bangladesh. A cross sectional web based survey. 

Dear Dr. Chomon:

I'm pleased to inform you that your manuscript has been deemed suitable for publication in PLOS ONE. Congratulations! Your manuscript is now with our production department. 

Kind regards, 

on behalf of

Dr. Sanjay Kumar Singh Patel 

Academic Editor

PLOS ONE